# Neuromotor changes in participants with a concussion history can be detected with a custom smartphone app

Christopher K. Rhea[1,2]*, Masahiro Yamada[1,3], Nikita A. Kuznetsov[1,4], Jason T. Jakiela[1,5], Chanel T. LoJacono[1,6], Scott E. Ross[1], F. J. Haran[1,7], Jason M. Bailie[8,9,10], W. Geoffrey Wright[11]

1 Department of Kinesiology, University of North Carolina at Greensboro, Greensboro, North Carolina, United States of America, 2 College of Health Sciences, Old Dominion University, Norfolk, Virginia, United States of America, 3 Moss Rehabilitation Research Institute, Elkins Park, Pennsylvania, United States of America, 4 Department of Psychology, University of Cincinnati, Cincinnati, Ohio, United States of America, 5 Department of Physical Therapy, University of Delaware, Newark, Delaware, United States of America, 6 Department of Kinesiology, Missouri Southern State University, Joplin, Missouri, United States of America, 7 Uniformed Services University of the Health Sciences, Bethesda, Maryland, United States of America, 8 Naval Hospital Camp Pendleton, Oceanside, California, United States of America, 9 Traumatic Brain Injury Center of Excellence (TBICoE), Silver Spring, Maryland, United States of America, 10 General Dynamics Information Technology, Falls Church, Virginia, United States of America, 11 Department of Health and Rehabilitation Sciences, Temple University, Philadelphia, Pennsylvania, United States of America

* crhea@odu.edu

**Data Availability Statement:** All relevant data are available within the paper and its Supporting Information files.

## Abstract

Neuromotor dysfunction after a concussion is common, but balance tests used to assess neuromotor dysfunction are typically subjective. Current objective balance tests are either cost- or space-prohibitive, or utilize a static balance protocol, which may mask neuromotor dysfunction due to the simplicity of the task. To address this gap, our team developed an Android-based smartphone app (portable and cost-effective) that uses the sensors in the device (objective) to record movement profiles during a stepping-in-place task (dynamic movement). The purpose of this study was to examine the extent to which our custom smartphone app and protocol could discriminate neuromotor behavior between concussed and non-concussed participants. Data were collected at two university laboratories and two military sites. Participants included civilians and Service Members (N = 216) with and without a clinically diagnosed concussion. Kinematic and variability metrics were derived from a thigh angle time series while the participants completed a series of stepping-in-place tasks in three conditions: eyes open, eyes closed, and head shake. We observed that the standard deviation of the mean maximum angular velocity of the thigh was higher in the participants with a concussion history in the eyes closed and head shake conditions of the stepping-in-place task. Consistent with the optimal movement variability hypothesis, we showed that increased movement variability occurs in participants with a concussion history, for which our smartphone app and protocol were sensitive enough to capture.

**Funding:** This work was supported by funding the Office of the Assistant Secretary of Defense for Health Affairs under award no. W81XWH-15-1-0094 to CKR. The sponsor did not play any role in the study design, data collection and analysis, decision to publish, or preparation of the manuscript. The opinions or assertions contained herein are the private ones of the authors and are not to be construed as official or reflecting the views of the Department of Defense, the Uniformed Services University of the Health Sciences or any other agency of the U.S. Government.

**Competing interests:** The authors have declared that no competing interests exist.

## Introduction

Concussions have been labeled as a public health concern due to our emerging knowledge of their short- and long-term negative effects [1], including dysfunction in working memory [2], altered sensitivity to sensory information [3], and balance control [4]. The latter has long been used as a clinical indicator of a potential concussion, as the head trauma event can lead to a neurological disruption, which in turn can affect downstream motor functions [5]. Thus, the integration of the neurological system with motor control (termed neuromotor control) affords an opportunity to probe motor behavior that may be disrupted from a concussion. Such was the impetus for balance tests commonly used in concussion assessment and follow-up care, such as the Balance Error Scoring System (BESS) [6], Sensory Organization Test (SOT) [7], and the Balance Tracking System (BTrackS) [8].

When selecting a balance test to be used for concussion assessment, it can either be subjective or objective [9]. Subjective tests, such as the BESS, rely on the perspective of the administrator to grade the participant's balance performance across various task conditions. While the BESS has been shown to have clinical utility [10], questions about its reliability have been discussed due to the subjective nature of the test [11]. A solution to this challenge is to use an objective test in which sensors and/or a computer quantifies balance control. Such is the case with the SOT, which uses a force plate to measure the center of pressure movement while the participant's visual, vestibular, and/or somatosensory information is altered in six conditions [7]. Although it is objective, the SOT has the challenge of being cost- and space-prohibitive in most clinical settings. The BTrackS is a more affordable way to objectively measure balance, consisting of a portable force plate connected to an iPad or laptop [8]. Although BtrackS may be suitable for a hospital/rehabilitation setting, it still requires a computer or tablet, which could limit its field deployment capabilities. A method to overcome this limitation can be using a smartphone app (Sway Medical, LLC, Tulsa, OK) [12–14]. This method can surpass the above-mentioned limitations (space, cost, portability, objective assessment).

A potential limitation of the existing portable technologies (i.e., BtrackS and Sway Medical) is that they were designed to probe neuromotor control via a static balance task. While such tests do have clinical utility [15, 16], a more challenging dynamic balance task could uncover dysfunctional behavior that may not have been observed in the easier static balance task. This postulate was explored in a recent review in which the authors examined the use of Fitts' Law as a way to control for and/or scale up task difficulty in balance tasks [17]. While they noted most of the papers in their review only included healthy populations, they did note that Fitts' law has been used in this context in clinical settings. The idea of scaling up task challenge (or difficulty) to discriminate neuromotor performance is the basis of many clinical tests, including the BESS test. In the original BESS paper where individuals with a concussion were assessed [6], the stances that presented the least difficulty (i.e., controlling balance on a firm surface) exhibited no differences between the concussed and non-concussed groups. However, once task difficulty was scaled up via balance control on a foam surface, discrimination between the groups was observed. Furthermore, a review of gait task analysis for concussions revealed that a more complex task showed differences in many more variables between non-concussed and concussed individuals than a simple gait task [18]. Lastly, in postural control, static balance performance may not reflect performance on a dynamic balance task [19], so they should not be used interchangeably. Thus, there is a need for a portable, objective, and dynamic assessment. To meet this challenge, our team developed an Android-based smartphone app (portable and cost-effective) that uses the sensors in the device (objective) to record movement profiles during a stepping-in-place task (dynamic movement). Version 1 of our app exhibited clinical utility by identifying neuromotor dysfunction in Service Members after

blast-exposure [20]. We then developed Version 2 of our app and published its reliability and validity [21].

The theoretical foundation for this line of research is rooted in the observation that variability in movement profiles is informative about the health of the neuromotor system. Unlike the traditional view about variability as noise, it has been proposed that movement variability provides sensitive measures of pathologies and movement disorders [22–30]—a position that has been formalized into the optimal movement variability hypothesis [22]. Previous research using this and related frameworks have shown that healthy biological systems exhibit a particular magnitude and structure of variability, while injury, aging, or disease can lead to a behavioral shift in movement such that it becomes more variable [23–29]. Relative to head trauma, previous research has shown changes in movement variability following a concussion [31, 32], which suggests that neuromotor dysfunction from a concussion can be tested via variability assessment. Moreover, using a metronome to anchor a timing task and then removing the metronome to examine the timing of the continuation movement (known as the synchronization–continuation paradigm) has a long history in basic motor behavior research [33, 34], along with concussion research [35]. This paradigm affords the ability to look at the response (movement pattern) to a constraint (metronome timing), with the working hypothesis that a neurological insult makes it more difficult to adhere to the timing constraint. Thus, we merged the optimal movement variability hypothesis with the synchronization-continuation paradigm to develop our smartphone dynamic balance protocol in order to create a theory-based and empirically driven approach to probe neuromotor performance after a concussion.

The purpose of the current study was to examine the extent to which the neuromotor behavior data obtained from our smartphone app could associate with a history of concussion. Since concussions are a problem in both civilian [36] and Service Member [37] populations, both groups were recruited for this study. We hypothesized that concussed participants would exhibit neuromotor dysfunction indexed by greater variability in their movement during the dynamic balance task.

## Materials and methods

### Participants

A total of 216 participants voluntarily participated in the study (N = 100 healthy civilians, N = 54 healthy Service Members, N = 44 civilians with a concussion history, N = 18 Serviced Members with a concussion history). All participants provided written informed consent. The demographics of N = 54 healthy Service Members were not obtained. Sex of the participants for whom demographics were obtained are as follows: (1) for the 100 healthy civilians, 51 were female and 49 were male, (2) for the 44 civilians with a concussion history, 26 were female and 18 were male, (3) and for the 18 Service Members with a concussion history, 4 were female and 14 were male. The height and weight of participants for whom demographics were available were similar between groups (Table 1). Civilian participants were recruited from two universities in the United States (one in the northeast and one in the southeast). Active-duty Service Members were recruited from two sites in the western United States. Both healthy (non-concussed) and participants with a concussion history were recruited. The sampling strategy at all sites was similar and consisted of a convenience sample from participants who self-identified as qualifying for the study. At the civilian sites, flyers were posted in public areas and information was provided verbally to various groups (e.g., via announcements in university courses and with student-athletes). At the military sites, participants were given the opportunity to volunteer outside of their standard daily duty. Participants in the concussion history group must have received an official diagnosis with a concussion by a medical professional in

**Table 1. Demographics by group (healthy, concussed) and population (civilian, Service Member) (M ± SD).**

| Demographics by Group (Civilians and Service Members Collapsed) | | | |
|---|---|---|---|
| Category | Healthy (n = 154) | Concussion History (n = 62) | p-value |
| Height (cm) | 171.99 ± 9.70 | 171.43 ± 12.59 | 0.749 |
| Weight (kg) | 70.93 ± 13.82 | 75.26 ± 16.19 | 0.072 |
| Time from concussion to testing (weeks) | N/A | 10.13 ± 8.67 | N/A |
| **Healthy and Concussion History within Each Population** | | | |
| Population | Civilians | | Service Members | |
| Category | Healthy (n = 100) | Concussion History (n = 44) | Healthy (n = 54) | Concussion History (n = 18) |
| Height (cm) | 171.99 ± 9.70 | 170.08 ± 13.87 | N/A | 174.72 ± 8.13 |
| Weight (kg) | 70.93 ± 13.82 | 73.75 ± 17.90 | N/A | 78.94 ± 13.46 |
| Time from concussion to testing (weeks) | N/A | 6.52 ± 6.32 | N/A | 18.16 ± 8.15 |

the past six months to qualify for our study. Inclusion criteria for both groups included: (1) 18–50 years old, (2) at least moderately active (> 3 times of physical activity per week), (3) BMI under 33, (4) non-smoker, (5) normal or corrected-to-normal vision, (6) no surgeries in the past six months, (7) ability to walk with a prosthetic or assistive device, and (8) not pregnant at the time of study data collection. The protocol was approved by two institutional review boards, and all participants signed the informed consent forms. If participants did not have a history of a concussion, they were allocated to the non-concussed healthy control (hereafter, healthy) group. If participants had a history of a concussion, they were assigned to the concussion history group. History of concussion was initially self-reported by the participants at the time of recruitment. However, prior to the day of testing, the participants medical professional provided a medical clearance consent form verifying that their patient did have a concussion within the last 6 months and that they supported them enrolling in our study with the knowledge of the type of testing that would occur. It is important to note that given the duration of time since the concussion event presented in Table 1, our population on average was not an acutely concussed population. Rather, our population was comprised of those who had a clinically diagnosed concussion within the past 6 months, and therefore are labeled as those with a concussion history.

## Study design and protocol

This study was a cross-sectional design that was part of a larger project that required participants to complete several static and dynamic balance tests in a single testing session. This paper focuses on our dynamic balance test from our smartphone app (named AccWalker). The protocol for AccWalker included strapping a customized armband onto the lateral aspect of the participants' thigh, halfway between the lateral epicondyle and the greater trochanter, and placing the smartphone in the armband holder (Fig 1A). Participants were instructed to (a) naturally step in place while synchronizing their step timing to the app's metronome, (b) look straightforward during trials, and (c) during familiarization trials, participants were informed to raise their knees higher if they were not moving as they normally walked (e.g., feet are not leaving the floor). The pacing metronome (period = 0.575 s or 1.15 s per stride) was provided for the initial 10 s of each trial. After 10 s, the metronome turned off, and the participant was instructed to continue stepping for 60 s. No other information was provided to participants in an attempt to capture natural kinematic patterns. The general protocol of the stepping-in-place task consisted of three trials for each of the three following conditions to alter sensory input: (1) with eyes open (EO), (2) with eyes closed (EC) to remove visual information, and (3) while rotating the

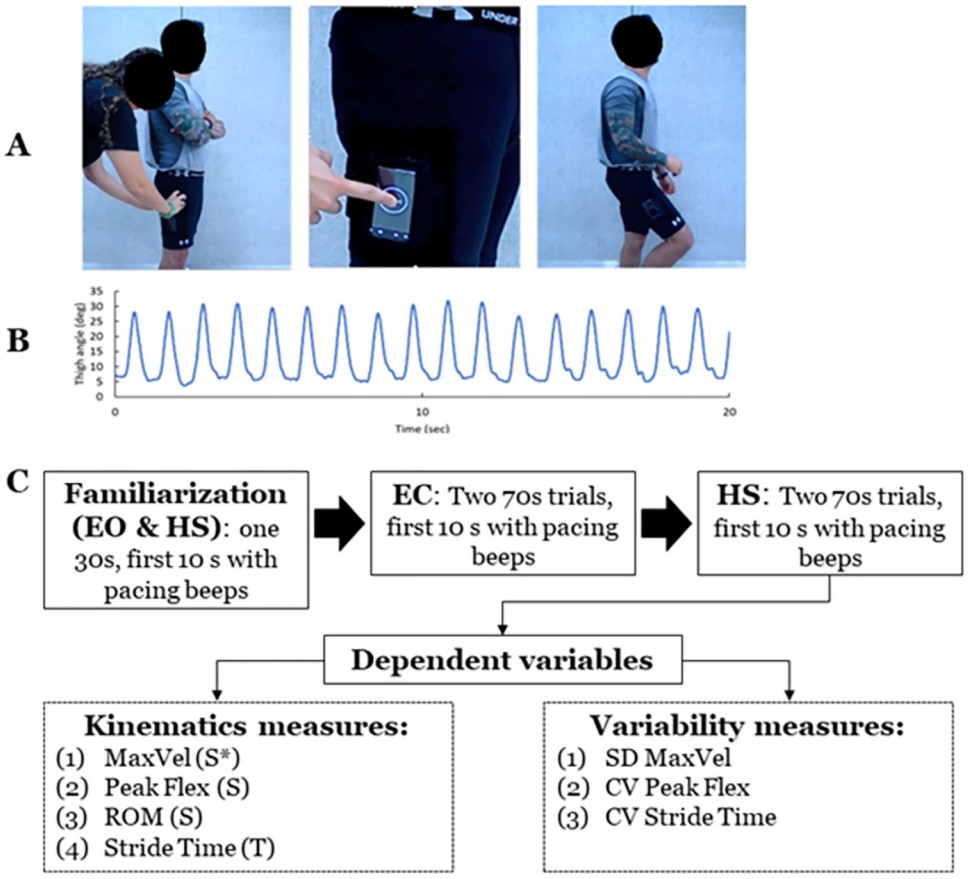

**Fig 1. Smartphone app.** (A) Placement of the phone on the thigh and the illustration of stepping movement. (B) Representative time series of the thigh flexion angle in the sagittal plane during the stepping in place task. (C) Study design and dependent variables extracted from the smartphone app.

head left-right to perturb vestibular information (head shake; HS). A 30-second practice session was provided for the EO and HS conditions prior to data collection to familiarize the participant with the task. The manual of the protocol was recorded by the smartphone app developing team for consistency and shared the protocols with partner teams.

For the present study, participants at the civilian sites performed 3 trials per condition for the EO, EC, and HS conditions (i.e., 60 seconds per trial x 3 trials x 3 conditions). Due to time constraints, participants at one military site performed only 2 trials of all three conditions (i.e., 60 seconds per trial x 2 trials x 3 conditions), and participants at the other military site performed 2 trials per condition only the EC and HS conditions (i.e., 60 second per trial x 2 trials x 2 conditions). In both sites, all participants completed the task in the order of EO, EC, HS for civilians and EC, HS for service members, with brief standing rest between trials (20–30 seconds). To compare common data that were collected at all sites, the mean of all dependent variables for the first 2 trials (instead of all 3 trials) were calculated for the EC and HS conditions, which were then submitted for statistical analysis. This is congruent with previous work showing acceptable concurrent validity when averaging 2 trials in our protocol [21].

## Data collection and dependent variables

With the phone placed on the lateral aspect of the thigh, the thigh flexion angle in the sagittal plane was quantified over time, with 0 degrees representing the leg perpendicular to the

ground plane (neutral standing position), positive angles representing thigh flexion angles beyond neutral, and negative angles representing thigh extension angles beyond neutral (Fig 1B). This thigh angle time series data were obtained from the Android smartphone's (Moto X2nd Generation, version 5.1 or Google Pixel, version 8.1.0 or 9) sensor fusion from the accelerometer, gyroscope, and magnetometer, which were recorded at 100.86Hz as described in our previous work [21]. The extracted thigh angle data from the phone were processed with a customized MATLAB script (R2020a, MathWorks Inc., Natick, MA). The obtained data were evenly distributed to 100Hz to consider sampling lags with spline interpolation. Then, the data were filtered with the 4$^{th}$ order low-pass Butterworth filter (both directions) with a cutoff at 5Hz. Non-normal trials (e.g., the phone slipped from the original position; participants stopped or paused during trials; data with noise) were visually detected. For all trials, the first 10 seconds from each trial, during the metronome tones were provided, were removed, resulting in the analysis of 60 s duration for each trial.

Our previous work showed which spatial and temporal variables derived from the thigh angle time series are reliable and valid [21]. For the present study, we report seven variables derived from the app (Fig 1C): four that describe general performance kinematics and three that describe movement variability. The general performance kinematics variables include (1) the mean of maximum angular velocity (MaxVel), (2) the mean of peak flexion angle (Peak-Flex), (3) the mean of knee range of motion (ROM), and (4) the mean of stride time. The first three variables represent spatial characteristics (i.e., primary variables) and the last variable represents the temporal characteristics (i.e., confirmation purpose). For MaxVel and PeakFlex, maximum values (peaks) of the knee angular velocity and knee angle, respectively, were obtained from each stride across one trial. The mean of these peaks was calculated. For the knee extension angle used in the ROM calculation, the obtained thigh angle data were partitioned into 100 bins and plotted into a histogram. Then, the most frequent bin was selected since the longest phase of a step-in-place task is the single-leg stance (i.e., when the knee is extended). The maximum value from this bin was defined as the knee extension angle. ROM was found as the mean difference between the peak flexion and knee extension. Stride time was determined by quantifying the intervals in seconds between maximal thigh flexion events. Stride time was analyzed to examine whether participants complied with the temporal restriction prescribed during the first 10 seconds of the task.

The variables that quantify movement variability are (1) SD MaxVel, (2) CV Peak Flex, and (3) CV Stride Time. To calculate SD MaxVel, the SD of MaxVel was extracted for each individual. Then, the mean and dispersion of the MaxVel SDs were calculated within each group. Thus, SD MaxVel represents the dispersion of SD (i.e., variability across individuals). For CV Peak Flex and CV Stride Time, CV was obtained by SD divided by its corresponding mean.

During the dependent variable extraction, non-normal trials were removed. Trials were considered as errors and removed if the phone slipped from the strap, participants did not comply with the metronome pace (e.g., participants stopped or paused during trials), or ROM was less than 10 degrees.

## Statistical design

For each dependent variable, a linear mixed-effect model analysis was adopted. The original model specification was three fixed effects and one random effect. The condition (EC, HS), group (Healthy, Concussion History), and the interaction between condition and group were fixed effects and the individual intercept variance was the random effect (Fig 2B). The model specification process was predetermined as follows: (a) the original model was tested, if necessary, with a model with different variance structures; (b) the condition x group interaction

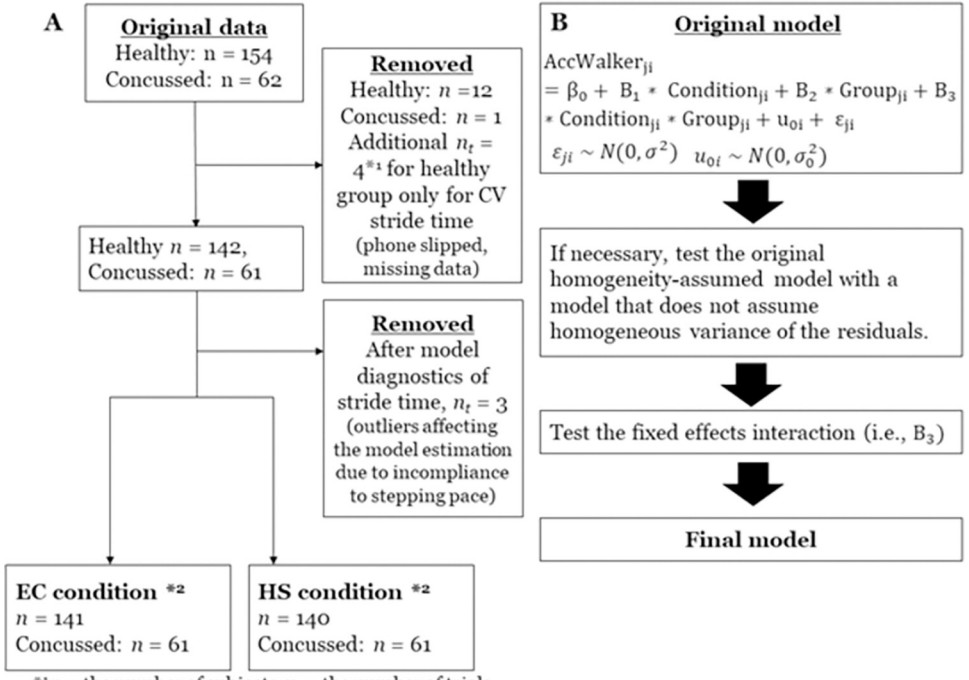

**Fig 2. Data processing and model specification flow chart.** Abbreviations: HS = head shake, EC = Eyes-closed conditions. (A) Flow chart of the data processing. The two boxes in the bottom are the sample size submitted for the statistical analyses for all variables except CV Stride time. [*1] n = the number of subjects, $n_t$ = the number of trials. [*2] for CV stride time, n = 138 healthy and n = 61 concussed participants for the EC condition and n = 141 healthy and n = 60 concussed participants for the HS condition were submitted for analyses. (B) the model specification process: Fixed effects coefficients are $B_0$, $B_1$, $B_2$, and $B_3$, j = j-th group of i-th individual. $u_{0i}$ represents the random effect of the individual intercept, and $e_{0i}$ represents the residuals, where both are assumed to be normally distributed.

term was omitted if the interaction term was not significant; and (c) the final model was determined based on the results of model comparison with Likelihood Ratio Test and AIC (Akaike Information Criterion) (Fig 2A). A model with parsimonious parameters was chosen if the two models were not statistically different. Since the task used in the present study constrained the temporal characteristics (i.e., stride time) and measured spatial characteristics, Stride Time was fitted first. The healthy group was a reference for the group variable, and the EC condition was a reference for the condition variable. The results were interpreted based on the output of the final model, with the coefficient value representing the magnitude change between the conditions or groups. Alpha was set at 0.05.

## Results

Participant demographics and time since concussion are presented in Table 1. No difference in height ($t_{160} = 0.320$, p = 0.749) and weight ($t_{160} = -1.812$, p = 0.072) were observed between healthy and concussion history groups.

During data processing, a total of 12 participants from the healthy group and one participant from the concussion history group were removed (Fig 2A). Nine of these participants were from the same cohort (*i.e.*, healthy service members). Further, additional four trials were removed for CV Stride time as outliers. Following the model specification for the Stride Time variable, three additional outliers were identified, which were likely due to the incompliance with the stepping pace. Thus, these trials were removed. This resulted in n = 141 healthy and

**Table 2. Final model and statistical results of each general performance kinematic dependent variable.**

| Stride Time | | | | | | |
|---|---|---|---|---|---|---|
| **Variable** | **Coefficient** | **SE** | **DF** | **t-value** | **p-value** | **Final model** |
| Intercept | 1.1503 | 0.006 | 200 | 208.716 | < 0.001 | $StrideTime_{ji} = \beta_0 + B_1 * Con_{ji} + B_2 * Grp_{ji} + + u_{0i} + \varepsilon_{ji}$ |
| Condition | -0.005 | 0.003 | 200 | -1.703 | 0.0901 | |
| Group | -0.017 | 0.001 | 200 | -1.718 | 0.0873 | |
| Mean Maximum Knee Flexion Velocity (Max Vel) in deg/s | | | | | | |
| Variable | Coefficient | SE | DF | t-value | p-value | Final model |
| Intercept | 170.314 | 3.309 | 200 | 51.476 | < 0.001 | $MaxVel_{ji} = \beta_0 + B_1 * Con_{ji} + B_2 * Grp_{ji} + + u_{0i} + \varepsilon_{ji}$ |
| Condition | -5.944 | 1.565 | 200 | -3.798 | 0.0002 | |
| Group | 8.301 | 5.847 | 200 | 1.421 | 0.157 | |
| Mean Peak Knee Flexion Angle (Peak Flex) in deg | | | | | | |
| Variable | Coefficient | SE | DF | t-value | p-value | Final model |
| Intercept | 41.663 | 0.842 | 200 | 49.472 | < 0.001 | $PeakFlex_{ji} = \beta_0 + B_1 * Con_{ji} + B_2 * Grp_{ji} + + u_{0i} + \varepsilon_{ji}$ |
| Condition | -1.639 | 0.379 | 200 | -4.326 | < 0.001 | |
| Group | 0.810 | 1.492 | 200 | 0.543 | 0.588 | |
| Mean of Range of Motion (ROM) in deg | | | | | | |
| Variable | Coefficient | SE | DF | t-value | p-value | Final model |
| Intercept | 37.782 | 0.782 | 200 | 48.323 | < 0.001 | $ROM_{ji} = \beta_0 + B_1 * Con_{ji} + B_2 * Grp_{ji} + + u_{0i} + \varepsilon_{ji}$ |
| Condition | -1.502 | 0.381 | 200 | -3.941 | 0.0001 | |
| Group | 1.261 | 1.379 | 200 | 0.914 | 0.362 | |

n = 61 concussion history participants for the EC condition, n = 140 healthy, and n = 61 concussion history participants for the HS condition (four additional trials removed for CV Stride Time) (Fig 2A). These remaining data were submitted for statistical analyses.

## General performance kinematics

The results of the final model are for the general performance kinematics are summarized in Table 2. As described above, we identified outliers that were likely due to incompliance with the stepping pace after the diagnosis of residuals of the final model. Thus, the model specification process was repeated without these outliers. First, homogeneity of variance between healthy and concussion history participants was potentially different from visual inspection. However, a model with a heterogeneous variance structure of the residuals was not statistically different from the original model with a homogeneous variance structure [$X^2(1) = 2.592$, p = 0.107]. Next, the group and condition interaction of the fixed effect was omitted in the final model since it was not significant [t = 1.001, p = 0.318]. The results of the final model (Table 2) showed no difference between conditions (p = 0.090) nor group (p = 0.087). The estimated means of the stride time were 1.15 s for the healthy group and 1.13 s for the concussion history group after controlling for the condition. This result suggests that the step interval of both groups approximated the target pace (i.e., 1.15 s), and the following spatial and variability dependent variables were not confounded by differences in the imposed temporal characteristics.

For other variables, the model specification process was similar to Stride time. Specifically, visual inspection showed an indication of different variances by group, but they were all non-significant, and no interaction was not found. Thus, the final model had two fixed effects (group and condition) and one random intercept. Therefore, the main effects of the final model were reported for other variables. For variability measures, we did not identify an

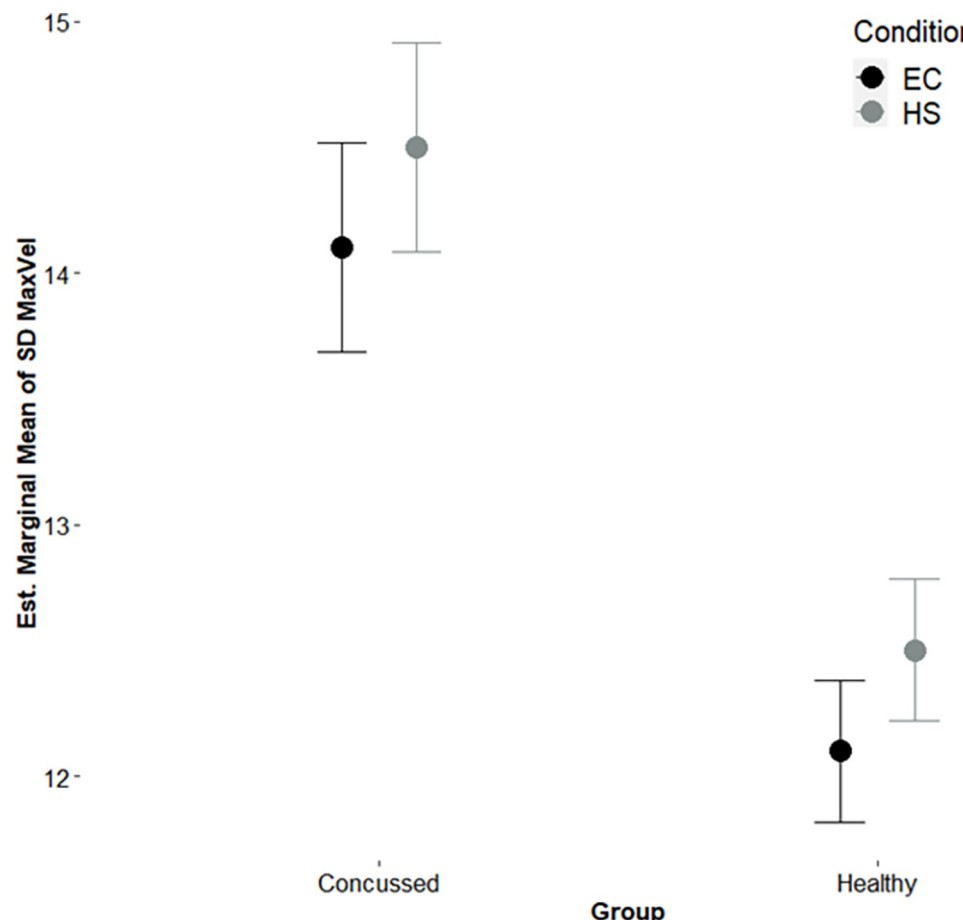

**Fig 3. Estimated marginal means of SD knee maximum angular velocity.** EC = Eyes-closed, HS = Head Shake conditions; Bars = Standard error of the mean; the values are estimated marginal means from the final model.

**Table 3. Final model and statistical results of each movement variability dependent variable.**

| Variable | Coefficient | SE | DF | t-value | p-value | Final model |
|---|---|---|---|---|---|---|
| **Variability (SD) of Maximum Velocity (SD_MaxVel)** | | | | | | |
| Intercept | 12.110 | 0.283 | 200 | 42.828 | < 0.001 | $SD\_MaxVel_{ji} = \beta_0 + B_1 * Con_{ji} + B_2 * Grp_{ji} + + u_{0i} + \varepsilon_{ji}$ |
| Condition | 0.400 | 0.185 | 200 | 2.167 | 0.0314 | |
| Group | 2.001 | 0.486 | 200 | 4.120 | < 0. 001 | |
| **Variability (CV) of Peak knee flexion (CV_PeakFlex)** | | | | | | |
| Intercept | 6.432 | 0.189 | 199 | 34.091 | < 0.001 | $CVPeakFlex_{ji} = \beta_0 + B_1 * Con_{ji} + B_2 * Grp_{ji} + + u_{0i} + \varepsilon_{ji}$ |
| Condition | 0.370 | 0.135 | 199 | 2.738 | 0.0067 | |
| Group | 0.268 | 0.319 | 199 | 0.839 | 0.4023 | |
| **Variability (CV) of Stride time (CV_Stride Time)** | | | | | | |
| Intercept | 2.510 | 0.085 | 200 | 29.356 | < 0.001 | $CV\_StrideTime_{ji} = \beta_0 + B_1 * Con_{ji} + B_2 * Grp_{ji} + + u_{0i} + \varepsilon_{ji}$ |
| Condition | 0.177 | 0.083 | 197 | 2.128 | 0.0346 | |
| Group | 0.241 | 0.135 | 200 | 1.785 | 0.0758 | |

indication of heterogeneous variance. Thus, after fitting the original model, it was compared with a model without an interaction term to determine the final model.

For the mean of maximum knee velocity (MaxVel), there was no main effect of group (p = 0.157), but the main effect was found in condition (p < 0.001), suggesting that the change from the EC condition to the HS condition was associated with 5.944 (m/s) decrease in Max-Vel (p < 0.001) (Fig 3).

For the mean of maximum knee flexion (PeakFlex), the results showed no significant effect of group (p = 0.588) and a significant main effect of condition, suggesting that the HS condition was associated with a 1.639-degrees decrease in the knee flexion angle relative to the EC condition (p < 0.001).

For the mean range of motion (ROM), similarly, there was no significant effect of group (p = 0.362) and a significant main effect of condition, suggesting that the HS condition was associated with a 1.502-degrees decrease in ROM relative to the EC condition (p < 0.001).

### Movement variability variables

The results of the final model are for the movement variability variables are summarized in Table 3. For the variability (SD) of MaxVel (SD_MaxVel), the main effects were found in both condition (p = 0.031) and group (p < 0.001) (Table 2). The change in the HS condition to the EC condition was associated with an increase in variability by 0.400. The concussion history group was associated with a 2.001-point variability increase after controlling for the condition.

For the variability (CV) of Peak Knee Flexion (CV_PeakFlex), the results of one participant were extreme scores. We were confident that it was a processing error (i.e., the extreme score was 34.92 compared to the estimated mean of 6.43 with SE = 0.189). Thus, this participant was removed. Our results showed no main effect of group (p = 0.402) with a main effect of condition (p = 0.007). The HS condition was associated with a 0.370-point variability increase compared to the EC condition.

For the variability (CV) of Stride Time (CV_Stride Time), the main effect was found in condition (p = 0.035) with no difference in group (p = 0.076). The HS condition was associated with an increase of variability by 0.177 points.

## Discussion

The purpose of the current study was to examine the extent to which the neuromotor behavior data obtained from our smartphone app could associate with a history of concussion. Aligned with our hypothesis, we showed that movement variability (via SD Max Vel) was significantly higher in the EC and HS conditions for participants with a concussion compared to participants without a concussion. We discuss our findings in the context of previous balance control tests for concussion assessment and using the optimal movement variability hypothesis to frame our observations.

Previous work has shown that neuromotor dysfunction occurs after a concussion [31, 38, 39]. Specifically, balance control becomes more variable due to neuromotor sensory integration challenges [32]. This is thought to result from the neurometabolic cascade that happens after a significant head impact [40], leading to challenges downstream in controlling motor actions [5]. For this reason, balance tasks have a long history of being used in concussion assessments [41]. Our findings align with previous work showing that the control of movement can become more variable after a concussion [31, 32, 42–44]. More specifically, our participants with a concussion exhibited a higher standard deviation of the maximum angular velocity of their thigh during the stepping-in-place task, indicating the inability to maintain consistent motion, even though they were cued by a metronome at the beginning of the task (utilizing the synchronization-continuation paradigm). Previous studies suggested that metrics

including velocity information are more sensitive to gait impairments in post-stroke patients than joint angles and timing parameters alone [45]. Our findings are consistent with this observation and extend it by adopting the paradigm in a manner suitable for field-testing.

The impetus for this project was the goal of creating an objective, cost-effective, and portable way to measure neuromotor control in individuals with a suspected concussion. Our solution for that challenge was to develop a custom smartphone app and a specific testing protocol. It should be noted that smartphone apps have been previously developed to meet this challenge, but many of these apps have not been rigorously and scientifically tested [46, 47]. An app that has been tested and is commercially available is the Sway Balance app (Sway Medical, LLC, Tulsa, OK), which has been shown to be valid [14], reliable [12], and have clinical utility [13]. Another empirically-based portable balance test is the BTrackS [8, 48, 49], which is an objective and relatively cost-effective solution to measure neuromotor performance [8]. In the context of concussion assessment, BTrackS was shown to have clinically acceptable sensitivity to detect dysfunctional balance performance [16]. In that study, 16 out of 25 participants with a concussion were observed to have increased variability in their balance control, an observation that aligns with our current findings. While the Sway Balance app and BTrackS have both demonstrated clinical utility, both assesses balance control with a static task, which may mask neurosensory dysfunction relative to a more challenging dynamic task [17, 18]. Thus, our dynamic balance protocol was developed to overcome this potential limitation in mind. However, it should be noted that the current paper does not address the static versus dynamic balance comparison. Nevertheless, our data do show that individuals with a concussion exhibited elevated variability in the dynamic balance control relative to non-concussed individuals, adding important insight to our knowledge base on ways that a concussion may affect movement patterns.

The observation that an increase in variability (SD maximum angular velocity) was exhibited in our participants with a history of concussion relative to our healthy participants aligns with the optimal movement variability hypothesis (OMVH) [22]. OMVH builds upon decades of research that show variability patterns in a biological system are informative about its health and functional capacity [22–30]. The original work in this area focused on a unidirectional change in the variability pattern—from a complex signal toward a less complex signal—in the presence of aging and disease [29]. This was articulated as the loss of complexity hypothesis, which was rooted in the premise that a healthy biological system exhibits a certain level of variability, which could be described mathematically as a more complex signal (commonly via the use of entropy metrics). When changes to the biological system occur due to aging or disease (e.g., reduced neural firing rate, reduced reaction time, reduced aerobic and anerobic capacity), a less variable (and less complex) signal was commonly observed. This framework originated in cardiovascular dynamics research [28, 29, 50, 51], but has since been extended to study human movement [22, 23, 27, 52–57]. Moreover, this framework has been extended to include a bi-directional movement of variability, which is the premise of the OMVH. That is, a continuum of variability is considered, with the "optimal" variability residing in the middle of the continuum. A shift away from the middle—toward less complex variability on one side or overly complex variability on the other side—represents a non-optimal variability pattern reflecting reduced functional capacity. In the context of this study and the dynamic balance task, the variability exhibited by the non-concussed individuals would be considered "optimal". The individuals with a concussion history exhibited elevated variability in their movement pattern relative to the non-concussed individuals. This observation aligns with the increased variability that was observed in the BESS [10] and in the SOT [38] after a concussion. Our findings also align with Purkayastha et al. [31], where they showed an increase in anterior-posterior variability during a static balance task in participants with a concussion relative to controls. While our study was not a within-subjects tracking study where data before and

after the concussive event were available, it would be hypothesized that the individuals with a concussion likely shifted from the optimal zone of variability toward the increased variability on the continuum. While outside the scope of this study, a prospective tracking study could confirm this shift, as well as the potential shift back toward the optimal variability zone after the concussion effects reside. Such a study design would help provide more contextual information when using the OMVH as the framework.

It should be noted that temporal dynamics have significantly contributed to our knowledge of the optimal movement variability hypothesis. Under this framework, temporal dynamics are typically examined via a metric(s) that quantify the structure of the variability (i.e., sample entropy). However, a limitation of this approach is that enough data points (and more importantly, cyclic events) must be present in order for repeating patterns to naturally emerge. Short datasets typically do not allow for an emergence of such repeating patterns, and thus are not good candidates for structure of variability metrics. Our task was confined to 60 seconds of stepping-in-place per trial, leading to ~52 steps on average. While nonlinear dynamics could be run on such a short dataset, it would open up questions about validity and reliability of such an approach. Thus, we elected to focus on magnitude of variability metrics (i.e., coefficient of variation), which is more appropriate to quantify variability in short time series.

While the current study was a retrospective design that tested participants with a concussion history, it is important to note that work has been done to predict the risk of motion sickness after head trauma [58–60]. This approach aligns with the general approach to identify risk factors for a concussion [61], which has led to questions related to what strategies can be used to modify the sport and/or player care to reduce concussion risk [62]. As these approaches advance, it is plausible that such predictive analytics could be part of a smartphone app similar to the one tested in this paper, which could enhance athlete care.

There are several limitations of this study to acknowledge. First, although the inclusion criteria required the concussive event to have occurred no longer than six months prior to our testing, the number of weeks since the event varied substantially in our population. Concussion symptoms can dissipate for many people 7–10 days after the event, so future work should explore the factor of time-since-injury as part of their study design. Prior to the main analysis, we had run a correlation analysis within the concussion history group between the dependent variable and time from concussion (n = 54; time from concussion was missing for n = 8). The results of the Spearman correlation did not show a significant association (rho = - 0.013, p = 0.888) (Fig 4). Thus, we did not include this variable in our model. However, it is possible that this characteristic is unique to our cohort. More research is warranted to understand movement variability and the time from concussion. Second, the testing environment differed between our groups (civilians in a university laboratory, Service Members with a concussion history in a military clinical setting, healthy Service Members on a military training site). While our app and protocol were designed to be deployed to diverse settings, future research should examine how different environmental factors (e.g., performing the task on concrete vs. gravel or in a quiet laboratory vs. noisy field-based setting) may influence the results. Third, although the stepping-in-place task is dynamic, one could argue that it is not functional outside of military context, so future work should focus on the inclusion of more functional tasks to examine dynamic balance control. Fourth, it is unclear if the dynamic measures in this study are more sensitive than existing static assessments, so a head-to-head study comparing static and dynamic balance assessments after a concussion is warranted in the future. Fifth, we acknowledge that sex can moderate the control of balance [63, 64], as well as the effect of a concussion on neuromotor control [65]. Unfortunately, sex was not reported for our entire dataset, so we were not able to include this variable in our statistical analyses. Thus, we recommend future work makes it a priority to include sex as a biological variable, congruent with

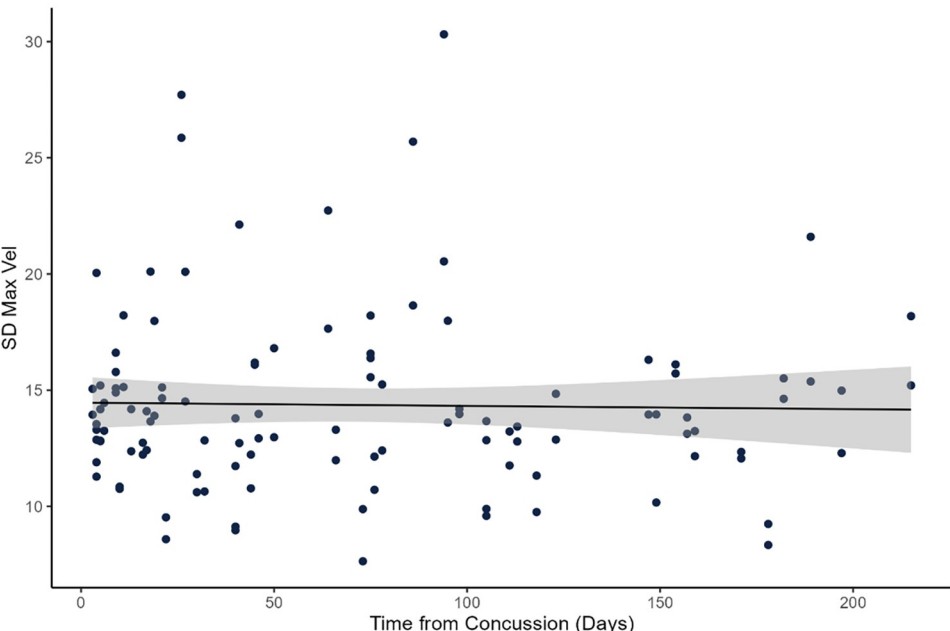

**Fig 4. Association between time from concussion and SD Max Vel.** Data showing no association between the number of days since the concussion event and neuromotor performance as assessed with SD Max Vel.

current NIH policy [66]. Sixth, the observation that SD MaxVel was sensitive to neuromotor changes, but CV PeakFlex and CV Stride Time were not (although generally trending in the same direction as SD MaxVel) needs to be further examined. Understanding this etiology will be key for future research. Lastly, the participants in this study could be generally characterized and young, fit, and physically active. Thus, these results may not generalize to populations outside of this characterization.

## Conclusion

We aimed to explore the extent to which neuromotor performance changes after a concussion could be detected with a custom smartphone app. We showed that increased variability in performance (SD maximum angular velocity of the thigh) during the dynamic balance task was observed in the participants with a concussion relative to the participants without a concussion. These findings are consistent with the optimal movement variability hypothesis that indicates variability characteristics will change if the neuromotor system is disrupted. Collectively, this study shows that neuromotor performance on an objective, cost-effective, and portable dynamic balance assessment is associated with a history of concussion.

## Supporting information

**S1 File. Raw data.** Dependent variable and demographic data for each participant. (XLSX)

## Author Contributions

**Conceptualization:** Christopher K. Rhea, Scott E. Ross, F. J. Haran, Jason M. Bailie, W. Geoffrey Wright.

**Data curation:** Christopher K. Rhea, Masahiro Yamada, Nikita A. Kuznetsov, Jason T. Jakiela, Chanel T. LoJacono, F. J. Haran, Jason M. Bailie, W. Geoffrey Wright.

**Formal analysis:** Christopher K. Rhea, Masahiro Yamada, Nikita A. Kuznetsov, Jason T. Jakiela, Chanel T. LoJacono, Scott E. Ross, F. J. Haran, Jason M. Bailie, W. Geoffrey Wright.

**Funding acquisition:** Christopher K. Rhea, Scott E. Ross, F. J. Haran, Jason M. Bailie, W. Geoffrey Wright.

**Investigation:** Christopher K. Rhea, Nikita A. Kuznetsov, Jason T. Jakiela, Chanel T. LoJacono, Scott E. Ross, Jason M. Bailie, W. Geoffrey Wright.

**Methodology:** Christopher K. Rhea, Nikita A. Kuznetsov, Jason T. Jakiela, Chanel T. LoJacono, Scott E. Ross, F. J. Haran, Jason M. Bailie, W. Geoffrey Wright.

**Project administration:** Christopher K. Rhea, Nikita A. Kuznetsov, Jason T. Jakiela, Chanel T. LoJacono, Scott E. Ross, Jason M. Bailie, W. Geoffrey Wright.

**Resources:** Christopher K. Rhea, Jason M. Bailie, W. Geoffrey Wright.

**Software:** Christopher K. Rhea, Nikita A. Kuznetsov.

**Supervision:** Christopher K. Rhea, Scott E. Ross, Jason M. Bailie, W. Geoffrey Wright.

**Validation:** Christopher K. Rhea, Scott E. Ross, W. Geoffrey Wright.

**Visualization:** Christopher K. Rhea.

**Writing – original draft:** Christopher K. Rhea, Masahiro Yamada.

**Writing – review & editing:** Christopher K. Rhea, Masahiro Yamada, Nikita A. Kuznetsov, Jason T. Jakiela, Chanel T. LoJacono, Scott E. Ross, F. J. Haran, Jason M. Bailie, W. Geoffrey Wright.

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
