## [Decision Letter · Decision Letter 0]

14 Apr 2022

PONE-D-22-03586Neuromotor changes after a concussion can be detected with a custom smartphone appPLOS ONE

Dear Dr. Rhea,

Thank you for submitting your manuscript to PLOS ONE. After careful consideration, we feel that it has merit but does not fully meet PLOS ONE’s publication criteria as it currently stands. Therefore, we invite you to submit a revised version of the manuscript that addresses the points raised during the review process.

Both Reviewers see merit in the work, but note significant problems with the submitted manuscript. Reviewer 2 recommends rejection, but I feel it may be possible to address the concerns raised in his/her review in a substantial revision. In revising, please pay careful attention to the issues raised by each of the Reviewers.

We look forward to receiving your revised manuscript.

Kind regards,

Thomas A Stoffregen, PhD

Academic Editor

PLOS ONE

Journal Requirements:

a) Did participants provide their written or verbal informed consent to participate in this study?

b) If consent was verbal, please explain i) why written consent was not obtained, ii) how you documented participant consent, and iii) whether the ethics committees/IRB approved this consent procedure

We will update your Data Availability statement on your behalf to reflect the information you provide."

4. We note that Figure 1 includes an image of a participant  in the study.

If you are unable to obtain consent from the subject of the photograph, you will need to remove the figure and any other textual identifying information or case descriptions for this individual

Reviewers' comments:

Reviewer's Responses to Questions

**Comments to the Author**

1. Is the manuscript technically sound, and do the data support the conclusions?

Reviewer #1: Partly

Reviewer #2: Partly

2. Has the statistical analysis been performed appropriately and rigorously? 

Reviewer #1: No

Reviewer #2: Yes

3. Have the authors made all data underlying the findings in their manuscript fully available?

Reviewer #1: Yes

Reviewer #2: Yes

4. Is the manuscript presented in an intelligible fashion and written in standard English?

Reviewer #1: Yes

Reviewer #2: Yes

5. Review Comments to the Author

Reviewer #1: The authors used a smartphone to measure the accelerative consequences of a stepping in place task that was applied to a large sample of adults, with vs. without recent concussion. One parameter of measured movement differed between groups. The authors propose their method as a cheap, effective way to obtain objective (i.e., non-observational) data on the movement consequences of concussion. The idea is excellent and the approach (generally) is good. With some important changes in analysis, the value of the study can be greatly increased.

It is necessary to report the numbers of women and men in your sample. It also is relevant: Posture and movement differ by sex (e.g., Kim et al., 2010), and aspects of concussion differ by sex (e.g., Alsalaheen et al., 2019). In revising, please conduct new analyses in which sex is an independent variable.

Much of the recent research relating concussion to quantitative measures of sway has focused on measures of temporal dynamics; something that is prominent in formulation and testing of the optimal movement variability hypothesis. The data obtained from the cell phones might be analyzed in any of a number of ways. In revising, please either 1) compute and include some measure of temporal dynamics or 2) provide an explicit motivation for not using the available data to assess temporal dynamics.

I understand that ANOVA was not appropriate, but the reversion to t-tests weakens the analysis. You can do better. Please consider conducting new analyses using mixed methods analysis (aka mixed effects modelling), which (in effect) is a form of ANOVA that is robust to the sorts of issues that exist in your data set (Brown & Prescott, 2006; Kreft & de Leeuw, 1998; Putt & Chinchilli, 1999).

The quantitative kinematics of the body are changed by concussion. However, recent evidence suggests that the quantitative kinematics of the body also may predict the risk of concussion in individuals (Chen et al., 2012, 2014, 2015). It would be helpful, in your Discussion, to address these effects. Might your cell phone approach be useful in a predictive context?

Alsalaheen, Johns, Bean, Almeida, Eckner, & Lorincz (2019). Women and men use different strategies to stabilize the head in response to impulsive loads: Implications for concussion injury risk. Journal of Orthopaedic & Sports Physical Therapy, 49, 779-786.

Brown H, Prescott R. Applied mixed models in medicine. New York: Wiley; 2006:215–44.

Chen, Y.-C., Hung, T.-H., Tseng, T.-C., Hsieh, C. C., Chen, F.-C., & Stoffregen, T. A. (2012). Pre-bout standing body sway differs between adult boxers who do and do not report post-bout motion sickness. PLOS ONE, 7(10): e46136. doi:10.1371/journal.pone.0046136.

Chen, Y.-C., Hung, T.-H., Tseng, T.-C., & Stoffregen, T. A. (2015). Postural precursors of post-boxing motion sickness in a manual aiming task. Ecological Psychology, 27, 26-42.

Chen, Y.-C., Tseng, T.-C., Hung, T.-H., & Stoffregen, T. A. (2014). Precursors of post-bout motion sickness in adolescent female boxers. Experimental Brain Research, 232, 2571-2579. DOI 10.1007/s00221-014-3910-4

Kim JW, Eom GM, Kim CS, Kim DH, Lee JH, Park BK, Hong J (2010) Sex differences in the postural sway characteristics of young and elderly subjects during quiet natural standing. Geriat Geront Int 10:191–198

Kreft, I. G., de Leeuw, J. Introducing multilevel modeling. NewYork: Sage Publications; 1998.

Putt M, Chinchilli VM. A mixed effects model for the analysis of repeated measures cross-over studies. Stat Med 1999; 18: 3037–58.

Reviewer #2: Thank you for the opportunity to review this manuscript. I commend the author for considering the need for novel and affordable methodologies which seek to identify neuromotor alterations in civilian and military individuals with a history of concussion. While the methodologies of the study are novel, the prepared manuscript is absent of pertinent information and critical data that are necessary for publication consideration. Thus, I recommend rejection for publication in PLOS ONE. Below I have summarized major concerns to support the authors with the next phases of their work.

1) The authors should consider reframing their manuscript to align with current literature regarding history of concussion. The authors refer to their participants as ‘concussed,’ which does not appear to be accurate given the length of time since last reported concussion. The participants’ current symptom status and whether they have been cleared to return to sport (or duty) should be included for reader interpretation. Likewise, I recommend authors include more detail on how history of concussion was quantified (e.g., was the ‘definition of concussion’ provided when asking participants to self-report? What about patients with a history of symptoms but no medical diagnosis?)

2) More clarification regarding the phone app as being an ‘objective outcome.’ While the smartphone app is objective, they are discriminating against a highly subjective diagnosis (concussion diagnosis varies greatly across medical professionals – was the criteria standardized?).

3) Clearer presentation of inclusion and exclusion criteria and demographics more generally (e.g., limb dominance, age, and entire group missing demographics). For instance, the inclusion of individuals with an ambulatory device or prosthetic would present a separate group of individuals with differences in gait or neuromotor control.

4) Lack of clarity in analyses. Specifically, the total n used in analyses differed without explanation (e.g., 154 healthy in table 1, but that number is not the n in any table 2 analysis).

5) While I appreciate the consideration of the OMVH, the data presented does not fully support the authors’ discussion points. The authors did not perform any nonlinear (e.g., complexity) analyses. Likewise, the optimal movement variability for healthy controls does not seem to be established in the literature.

In summary, while we do think the novelty of devices to measure neuromotor changes in the field are important to gain a greater understanding of individual deficits following concussive injury, the concerns stated above and other minor issues within the manuscript do not meet the current standards for publication in PLOS ONE.

6. PLOS authors have the option to publish the peer review history of their article (what does this mean?). If published, this will include your full peer review and any attached files.

Reviewer #1: No

Reviewer #2: No

---

## [Author Response · Author response to Decision Letter 0]

3 Jun 2022

The authors would like to thank the reviewers for their thoughtful comments, as well as thanking the editor for the opportunity to resubmit our paper for publication consideration in PLOS ONE. We believe the reviewers raised some significant and high-quality questions. We have attended to all of the reviewer’s comments/suggestions in the manuscript and/or in this document and have provided a point-by-point response to all comments/suggestions below. The reviewer’s comments are in bold, our responses are in normal typeface, and the text that has been added to the manuscript is in red (please see separate Response to Reviewers document).

REVIEWER #1 COMMENTS

 The authors used a smartphone to measure the accelerative consequences of a stepping in place task that was applied to a large sample of adults, with vs. without recent concussion. One parameter of measured movement differed between groups. The authors propose their method as a cheap, effective way to obtain objective (i.e., non-observational) data on the movement consequences of concussion. The idea is excellent and the approach (generally) is good. With some important changes in analysis, the value of the study can be greatly increased.

Thank you for your encouraging comments. We have heeded your and the other reviewer’s comments to strengthen the manuscript. 

 It is necessary to report the numbers of women and men in your sample. It also is relevant: Posture and movement differ by sex (e.g., Kim et al., 2010), and aspects of concussion differ by sex (e.g., Alsalaheen et al., 2019). In revising, please conduct new analyses in which sex is an independent variable.

We appreciate your comment, especially since we have focused our related work on the very sex differences you mention (Rhea et al., 2019). Unfortunately, we do not have sex data for all our participants. The reason for this is a large portion of our sample (all health Service Members, N=54) were collected as part of a larger previous study at a Department of Defense site and sex was not recorded as part of that dataset. Nevertheless, we agree that it is important to report sex for the participants for whom we have those data, so we have added the following text to lines 122-126:

“The demographics of N = 54 healthy Service Members were not obtained. Sex of the participants for whom demographics were obtained are as follows: (1) for the 100 healthy civilians, 51 were female and 49 were male, (2) for the 44 civilians with a concussion history, 26 were female and 18 were male, (3) and for the 18 Service Members with a concussion history, 4 were female and 14 were male.”

Since we do not have sex for all of our participants, we believe that re-running the analyses with sex as an independent variable would provide an impartial picture. Sex differences in the healthy civilians wouldn’t be that interesting, as it is the sex differences after a concussion that researchers typically focus on. With our relatively small and unbalanced concussed sample that would need to be further separated between Service Members and civilians, it is likely such sex-dependent analyses would be underpowered. A key feature of our study as presented is the large sample size (N=216 participants) and focusing on an analysis that splinters the groups to such small sub-samples would likely detract from this manuscript. However, we agree that your suggestion warrants serious consideration for future work. Accordingly, we have added the following text to lines 415-419:

“Fifth, we acknowledge that sex can moderate the control of balance [63, 64], as well as the effect of a concussion on neuromotor control [65]. Unfortunately, sex was not reported for our entire dataset, so we were not able to include this variable in our statistical analyses. Thus, we recommend future work makes it a priority to include sex as a biological variable, congruent with current NIH policy [66].”

Rhea, C.K., Schleich, K.N., Washington. L., Glass, S.M., Ross, S.E., Etnier, J.L., Wright, W.G., Goble, D.J., & Duffy, D.M. (2019). Neuromotor and neurocognitive performance in female American football players. Athletic Training & Sports Health Care, 11(5), 224-233. https://doi.org/10.3928/19425864-20181101-01

 Much of the recent research relating concussion to quantitative measures of sway has focused on measures of temporal dynamics; something that is prominent in formulation and testing of the optimal movement variability hypothesis. The data obtained from the cell phones might be analyzed in any of a number of ways. In revising, please either 1) compute and include some measure of temporal dynamics or 2) provide an explicit motivation for not using the available data to assess temporal dynamics.

Thank you for bringing up this important point. We agree with your comment that temporal dynamics have been shown to be important quantitative measures. We did include temporal metrics (i.e., stride time, variability of maximum velocity, variability of stride time), but we suspect your reference to temporal dynamics refers more to measures that quantify the evolution of the dynamics over time, such as nonlinear and/or complexity analyses (e.g., sample entropy). Due to our stepping-in-place task being confined to 60 seconds, such nonlinear analyses are not good candidates for our dataset. To acknowledge our data limitation, but with a nod to your point, we have added the following text to lines 385-394:

“It should be noted that temporal dynamics have significantly contributed to our knowledge of the optimal movement variability hypothesis. Under this framework, temporal dynamics are typically examined via a metric(s) that quantify the structure of the variability (i.e., sample entropy). However, a limitation of this approach is that enough data points (and more importantly, cyclic events) must be present in order for repeating patterns to naturally emerge. Short datasets typically do not allow for an emergence of such repeating patterns, and thus are not good candidates for structure of variability metrics. Our task was confined to 60 seconds of stepping-in-place per trial, leading to ~52 steps on average. While nonlinear dynamics could be run on such a short dataset, it would open up questions about validity and reliability of such an approach. Thus, we elected to focus on magnitude of variability metrics (i.e., coefficient of variation), which is more appropriate to quantify variability in short time series.”

 I understand that ANOVA was not appropriate, but the reversion to t-tests weakens the analysis. You can do better. Please consider conducting new analyses using mixed methods analysis (aka mixed effects modelling), which (in effect) is a form of ANOVA that is robust to the sorts of issues that exist in your data set (Brown & Prescott, 2006; Kreft & de Leeuw, 1998; Putt & Chinchilli, 1999).

Thank you for this consideration. To address your comment, we reanalyzed our data using a linear mixed effect model. Our revised statistical analysis is reported in lines 226-239:

“For each dependent variable, a linear mixed-effect model analysis was adopted. The original model specification was three fixed effects and one random effect. The condition (EC, HS), group (Healthy, Concussed), and the interaction between condition and group were fixed effects and the individual intercept variance was the random effect (Fig 2B). The model specification process was predetermined as follows: (a) the original model was tested, if necessary, with a model with different variance structures; (b) the condition x group interaction term was omitted if the interaction term was not significant; and (c) the final model was determined based on the results of model comparison with Likelihood Ratio Test and AIC (Akaike Information Criterion) (Fig 2A). A model with parsimonious parameters was chosen if the two models were not statistically different. Since the task used in the present study constrained the temporal characteristics (i.e., stride time) and measured spatial characteristics, Stride Time was fitted first. The healthy group was a reference for the group variable, and the EC condition was a reference for the condition variable. The results were interpreted based on the output of the final model, with the coefficient value representing the magnitude change between the conditions or groups. Alpha was set at 0.05.”

 Also, the analysis process has been revised and presented in a figure (see Fig 2B below).

Fig 2. Data processing and model specification flow chart. Abbreviations: HS = head shake, EC = Eyes-closed conditions. (A) Flow chart of the data processing. The two boxes in the bottom are the sample size submitted for the statistical analyses for all variables except CV Stride time *2 for CV stride time, n = 138 healthy and n = 61 concussed participants for the EC condition and n = 141 healthy and n = 60 concussed participants for the HS condition were submitted for analyses. (B) the model specification process: Fixed effects coefficients are B0, B1, B2, and B3, j = j-th group of i-th individual. u0i represents the random effect of the individual intercept, and e0i represents the residuals, where both are assumed to be normally distributed. 

Finally, we have added the new statistical results in Tables 2 and 3 to lines 293 and 309 (also pasted below). 

Stride Time 

Variable Coefficient SE DF t-value p-value Final model

Intercept 1.1503 0.006 200 208.716 < 0.001 StrideTime_ji=β_0+ B_1* Con_ji+B_2*Grp_ji++u_0i+ ε_ji

Condition -0.005 0.003 200 -1.703 0.0901 

Group -0.017 0.001 200 -1.718 0.0873 

Mean Maximum Knee Flexion Velocity (Max Vel) in deg/s

Variable Coefficient SE DF t-value p-value Final model

Intercept 170.314 3.309 200 51.476 < 0.001 MaxVel_ji=β_0+ B_1* Con_ji+B_2*Grp_ji++u_0i+ ε_ji

Condition -5.944 1.565 200 -3.798 0.0002 

Group 8.301 5.847 200 1.421 0.157 

Mean Peak Knee Flexion Angle (Peak Flex) in deg

Variable Coefficient SE DF t-value p-value Final model

Intercept 41.663 0.842 200 49.472 < 0.001 PeakFlex_ji=β_0+ B_1* Con_ji+B_2*Grp_ji++u_0i+ ε_ji

Condition -1.639 0.379 200 -4.326 < 0.001 

Group 0.810 1.492 200 0.543 0.588 

Mean of Range of Motion (ROM) in deg

Variable Coefficient SE DF t-value p-value Final model

Intercept 37.782 0.782 200 48.323 < 0.001 ROM_ji=β_0+ B_1* Con_ji+B_2*Grp_ji++u_0i+ ε_ji

Condition -1.502 0.381 200 -3.941 0.0001 

Group 1.261 1.379 200 0.914 0.362 

Table 2. Final model and statistical results of each general performance kinematic dependent variable. 

Variability (SD) of Maximum Veloicity (SD_MaxVel) 

Variable Coefficient SE DF t-value p-value Final model

Intercept 12.110 0.283 200 42,828 < 0.001 〖SD_MaxVel〗_ji=β_0+ B_1* Con_ji+B_2*Grp_ji++u_0i+ ε_ji

Condition 0.400 0.185 200 2.167 0.0314 

Group 2.001 0.486 200 4,120 0.0001 

Variability (CV) of Peak knee flexion (CV_PeakFlex)

Variable Coefficient SE DF t-value p-value Final model

Intercept 6.432 0.189 199 34.091 < 0.001 CVPeakFlex_ji=β_0+ B_1* Con_ji+B_2*Grp_ji++u_0i+ ε_ji

Condition 0.370 0.135 199 2.738 0.0067 

Group 0.268 0.319 199 0.839 0.4023 

Variability (CV) of Stride time (CV_Stride Time)

Variable Coefficient SE DF t-value p-value Final model

Intercept 2.510 0.085 200 29.356 < 0.001 〖CV_StrideTime〗_ji=β_0+ B_1* Con_ji+B_2*Grp_ji++u_0i+ ε_ji

Condition 0.177 0.083 197 2.128 0.0346 

Group 0.241 0.135 200 1.785 0.0758 

Table 3. Final model and statistical results of each movement variability dependent variable.

 The quantitative kinematics of the body are changed by concussion. However, recent evidence suggests that the quantitative kinematics of the body also may predict the risk of concussion in individuals (Chen et al., 2012, 2014, 2015). It would be helpful, in your Discussion, to address these effects. Might your cell phone approach be useful in a predictive context?

Thank you for connecting the predictive work in this space with our current effort. To address this connection, we have added the following text to lines 395-401:

“While the current study was a retrospective design that tested participants with a concussion history, it is important to note that work has been done to predict the risk of motion sickness after head trauma [58-60]. This approach aligns with the general approach to identify risk factors for a concussion [61], which has led to questions related to what strategies can be used to modify the sport and/or player care to reduce concussion risk [62]. As these approaches advance, it is plausible that such predictive analytics could be part of a smartphone app similar to the one tested in this paper, which could enhance athlete care.”

REVIEWER #2 COMMENTS

Thank you for the opportunity to review this manuscript. I commend the author for considering the need for novel and affordable methodologies which seek to identify neuromotor alterations in civilian and military individuals with a history of concussion. While the methodologies of the study are novel, the prepared manuscript is absent of pertinent information and critical data that are necessary for publication consideration. Thus, I recommend rejection for publication in PLOS ONE. Below I have summarized major concerns to support the authors with the next phases of their work.

Thank you for your critiques and comments. We have revised our manuscript based on your comments, especially by adding more pertinent information related to our study. 

 The authors should consider reframing their manuscript to align with current literature regarding history of concussion. The authors refer to their participants as ‘concussed,’ which does not appear to be accurate given the length of time since last reported concussion. The participants’ current symptom status and whether they have been cleared to return to sport (or duty) should be included for reader interpretation. Likewise, I recommend authors include more detail on how history of concussion was quantified (e.g., was the ‘definition of concussion’ provided when asking participants to self-report? What about patients with a history of symptoms but no medical diagnosis?)

We appreciate you brining up this important detail and we agree with your comments. Given the average number of weeks since the concussion event presented in Table 1, we agree that it is important to frame the results as those with a history of concussion, rather than a concussed group. To clarify group membership, the following information was added to lines 142-150:

“If participants had a history of a concussion, they were assigned to the concussion history group. History of concussion was initially self-reported by the participants at the time of recruitment. However, prior to the day of testing, the participants medical professional provided a medical clearance consent form verifying that their patient did have a concussion within the last 6 months and that they supported them enrolling in our study with the knowledge of the type of testing that would occur. It is important to note that given the duration of time since the concussion event presented in Table 1, our population on average was not an acutely concussed population. Rather, our population was comprised of those who had a clinically diagnosed concussion within the past 6 months, and therefore are labeled as those with a concussion history.”

Moreover, we have also changed our language that used “concussed” to “those with concussion history” in the Title, Abstract, Methods, Results, and Discussion sections as denoted by the red text throughout. 

Unfortunately, we do not have data on whether those with a concussion history were cleared to return to sport and active duty at the time of our testing. As for the symptoms, that dataset is incomplete. The Military Acute Concussion Evaluation (MACE)—which includes a symptom checklist—was given to the participants at the two civilian sites and the clinical military site. However, at one of the civilian sites, participants only completed the cognitive summary section of the MACE and did not fill out the symptom reporting section. Thus, we do not have symptom reporting for the civilians at that site (n=50 healthy and n=28 participants with concussion history), nor do we have symptom reporting at the military site where the healthy Service Members (n=54) participated, as the MACE was not given at that site. 

 More clarification regarding the phone app as being an ‘objective outcome.’ While the smartphone app is objective, they are discriminating against a highly subjective diagnosis (concussion diagnosis varies greatly across medical professionals – was the criteria standardized?).

That is a great point. The leading developer team created a manual video clip for other co-investigators. We also had meetings to ensure investigators at all sites understood the protocols. We only had a few criteria imposed on participants to capture natural stepping patterns. Additionally, there were no subjective ratings or diagnoses involved in our protocols. Thus, we believe that differences in the variables from the smartphone app can be reasonably considered as objective measures relative to the gold standards of concussion assessment (e.g., clinical assessment and criteria/symptom-checking surveys/diagnosis). We have revised our manuscript to include this information in lines 158-182. 

“The protocol for AccWalker included strapping a customized armband onto the lateral aspect of the participants’ thigh, halfway between the lateral epicondyle and the greater trochanter, and placing the smartphone in the armband holder (Fig 1A). Participants were instructed to (a) naturally step in place while synchronizing their step timing to the app’s metronome, (b) look straightforward during trials, and (c) during familiarization trials, participants were informed to raise their knees higher if they were not moving as they normally walked (e.g., feet are not leaving the floor). The pacing metronome (period = 0.575 s or 1.15 s per stride) was provided for the initial 10 s of each trial. After 10 s, the metronome turned off, and the participant was instructed to continue stepping for 60 s. No other information was provided to participants in an attempt to capture natural kinematic patterns. The general protocol of the stepping-in-place task consisted of three trials for each of the three following conditions to alter sensory input: (1) with eyes open (EO), (2) with eyes closed (EC) to remove visual information, and (3) while rotating the head left-right to perturb vestibular information (head shake; HS). A 30-second practice session was provided for the EO and HS conditions prior to data collection to familiarize the participant with the task. The manual of the protocol was recorded by the smartphone app developing team for consistency and shared the protocols with partner teams. 

For the present study, participants at the civilian sites performed 3 trials per condition for the EO, EC, and HS conditions (i.e., 60 seconds per trial x 3 trials x 3 conditions). Due to time constraints, participants at one military site performed only 2 trials of all three conditions (i.e., 60 seconds per trial x 2 trials x 3 conditions), and participants at the other military site performed 2 trials per condition only the EC and HS conditions (i.e., 60 second per trial x 2 trials x 2 conditions). In both sites, all participants completed the task in the order of EO, EC, HS for civilians and EC, HS for service members, with brief standing rest between trials (20 - 30 seconds). To compare common data that were collected at all sites, the mean of all dependent variables for 2 trials (instead of 3 trials) were calculated for the EC and HS conditions, which were then submitted for statistical analysis. This is congruent with previous work showing acceptable concurrent validity when averaging 2 trials in our protocol [21].”

 Clearer presentation of inclusion and exclusion criteria and demographics more generally (e.g., limb dominance, age, and entire group missing demographics). For instance, the inclusion of individuals with an ambulatory device or prosthetic would present a separate group of individuals with differences in gait or neuromotor control.

The only inclusion criteria used for our study are listed in lines 136-139:

“Inclusion criteria for both groups included: (1) 18-50 years old, (2) at least moderately active (> 3 times of physical activity per week), (3) BMI under 33, (4) non-smoker, (5) normal or corrected-to-normal vision, (6) no surgeries in the past six months, (7) ability to walk with a prosthetic or assistive device, and (8) not pregnant at the time of study data collection.”

Unfortunately, we do not have sex data for all our participants. The reason for this is a large portion of our sample (all health Service Members, N=54) were collected as part of a larger previous study at a Department of Defense site and sex was not recorded as part of that dataset (nor were their demographics collected at that site). Nevertheless, we believe that it is important to report sex for the participants for whom we have those data, so we have added the following text to lines 122-126:

“The demographics of N = 54 healthy Service Members were not obtained. Sex of the participants for whom demographics were obtained are as follows: (1) for the 100 healthy civilians, 51 were female and 49 were male, (2) for the 44 civilians with a concussion history, 26 were female and 18 were male, (3) and for the 18 Service Members with a concussion history, 4 were female and 14 were male.”

We did not collect information on limb dominance. Any individuals with an ambulatory device or prosethic would have been excluded based on the inclusion criteria #7 above. 

 Lack of clarity in analyses. Specifically, the total n used in analyses differed without explanation (e.g., 154 healthy in table 1, but that number is not the n in any table 2 analysis).

Thank you for the opportunity to clarify the differences in the sample size. We created a flow chart to clarify the analysis procedure (Figure 2). 

Fig 2. Data processing and model specification flow chart. Abbreviations: HS = head shake, EC = Eyes-closed conditions. (A) Flow chart of the data processing. The two boxes in the bottom are the sample size submitted for the statistical analyses for all variables except CV Stride time *2 for CV stride time, n = 138 healthy and n = 61 concussed participants for the EC condition and n = 141 healthy and n = 60 concussed participants for the HS condition were submitted for analyses. (B) the model specification process: Fixed effects coefficients are B0, B1, B2, and B3, j = j-th group of i-th individual. u0i represents the random effect of the individual intercept, and e0i represents the residuals, where both are assumed to be normally distributed. 

 While I appreciate the consideration of the OMVH, the data presented does not fully support the authors’ discussion points. The authors did not perform any nonlinear (e.g., complexity) analyses. Likewise, the optimal movement variability for healthy controls does not seem to be established in the literature.

Thank you for bringing up this important point. We believe that the OMVH encompasses various ‘families’ of variability, not necessarily derived from the chaos, fractal, or synergy theories, but also theories related to the magnitude of variability (e.g., optimal control theory, UCM, Bernstein’s concept, dynamical system theory). Moreover, we suggest that the data derived from our task is not a good candidate for nonlinear analyses. To acknowledge our data limitation, but with a nod to your point, we have added the following text to lines 385-394:

“It should be noted that temporal dynamics have significantly contributed to our knowledge of the optimal movement variability hypothesis. Under this framework, temporal dynamics are typically examined via a metric(s) that quantify the structure of the variability (i.e., sample entropy). However, a limitation of this approach is that enough data points (and more importantly, cyclic events) must be present in order for repeating patterns to naturally emerge. Short datasets typically do not allow for an emergence of such repeating patterns, and thus are not good candidates for structure of variability metrics. Our task was confined to 60 seconds of stepping-in-place per trial, leading to ~52 steps on average. While nonlinear dynamics could be run on such a short dataset, it would open up questions about validity and reliability of such an approach. Thus, we elected to focus on magnitude of variability metrics (i.e., coefficient of variation), which is more appropriate to quantify variability in short time series.”

In summary, while we do think the novelty of devices to measure neuromotor changes in the field are important to gain a greater understanding of individual deficits following concussive injury, the concerns stated above and other minor issues within the manuscript do not meet the current standards for publication in PLOS ONE.

Thank you for pointing the limitations of the manuscript. We have revised our manuscript based on your comments, and we believe the revised manuscript meets the standards for publication in PLOS ONE.

---

## [Decision Letter · Decision Letter 1]

2 Oct 2022

PONE-D-22-03586R1Neuromotor changes in participants with a concussion history can be detected with a custom smartphone appPLOS ONE

Dear Dr. Rhea,

Thank you for submitting your manuscript to PLOS ONE. After careful consideration, we feel that it has merit but does not fully meet PLOS ONE’s publication criteria as it currently stands. Therefore, we invite you to submit a revised version of the manuscript that addresses the points raised during the review process.

We look forward to receiving your revised manuscript.

Kind regards,

Callam Davidson

Editorial Office

PLOS ONE

Journal Requirements:

Please define all abbreviations in Figure 1 (in the legend).

Line 260: Please correct ‘remained’ to ‘remaining’.

Table 3: Per our statistical reporting guidelines (https://journals.plos.org/plosone/s/submission-guidelines.#loc-statistical-reporting), please report exact p-values for all values greater than or equal to 0.001. P-values less than 0.001 may be expressed as p < 0.001, or as exponentials in studies of genetic associations.

Line 424: Please capitalise ‘We’.

Reviewers' comments:

Reviewer's Responses to Questions

**Comments to the Author**

1. If the authors have adequately addressed your comments raised in a previous round of review and you feel that this manuscript is now acceptable for publication, you may indicate that here to bypass the “Comments to the Author” section, enter your conflict of interest statement in the “Confidential to Editor” section, and submit your "Accept" recommendation.

Reviewer #1: All comments have been addressed

Reviewer #3: (No Response)

2. Is the manuscript technically sound, and do the data support the conclusions?

Reviewer #1: Yes

Reviewer #3: Partly

3. Has the statistical analysis been performed appropriately and rigorously? 

Reviewer #1: Yes

Reviewer #3: I Don't Know

4. Have the authors made all data underlying the findings in their manuscript fully available?

Reviewer #1: Yes

Reviewer #3: Yes

5. Is the manuscript presented in an intelligible fashion and written in standard English?

Reviewer #1: Yes

Reviewer #3: Yes

6. Review Comments to the Author

Reviewer #1: The revision is good. It should be published. here are some extra characters to meet the minimum character count that is required.

Reviewer #3: This is the first revision of a manuscript reporting the results of a study of military and civilian participants with and without a recent (6-18 weeks) history of concussion. The SD of the mean maximum angular velocity of the thigh was found to be higher among participants with concussion history during the eyes closed and head shake conditions, compared to those without concussion history.

Most importantly, this analysis shows a statistical association, not discriminative ability – to do that would require a classification workflow (e.g., held out test set, classification performance assessment). I would encourage clarification of the language throughout the manuscript on this point.

Although the authors are to be commended for sharing the raw data from the study, it would help the reader to see descriptive analyses of the study variables – for example, a tabulation of the central tendency and distribution _and_ a scatter plot of the SD of maximum velocity (and the other ~6 potential input variables to a discriminative tool), across concussion vs. no concussion groups.

Specific comments

Both civilians and service members were young (18-50), healthy, and fit (height/weight). These results may not generalize to different populations.

For subgroups among whom 3 trials were performed, only 2 were analyzed. How were the 2/3 selected?

Was participant history of musculoskeletal injury collected? Such injuries could affect both range of motion and balance.

Table 1 and Results (first paragraph) – it is generally useful to have key comparisons be shown between table columns, with p-values shown in the table rather than only in text.

The time from concussion to testing was quite different between the two populations (6+ weeks civilians, ~18 weeks military) in a way that could affect results. Statistical associations were not found, but a more standardized study would be necessary before any final conclusions were drawn.

7. PLOS authors have the option to publish the peer review history of their article (what does this mean?). If published, this will include your full peer review and any attached files.

Reviewer #1: No

Reviewer #3: **Yes: **Tellen D. Bennett

---

## [Author Response · Author response to Decision Letter 1]

14 Nov 2022

Please see the Response to Reviewers attached document.

---

## [Decision Letter · Decision Letter 2]

29 Nov 2022

Neuromotor changes in participants with a concussion history can be detected with a custom smartphone app

PONE-D-22-03586R2

Dear Dr. Rhea,

We’re pleased to inform you that your manuscript has been judged scientifically suitable for publication and will be formally accepted for publication once it meets all outstanding technical requirements.

Kind regards,

Ryan Thomas Roemmich

Academic Editor

PLOS ONE

Additional Editor Comments (optional):

Reviewers' comments:

Reviewer's Responses to Questions

**Comments to the Author**

1. If the authors have adequately addressed your comments raised in a previous round of review and you feel that this manuscript is now acceptable for publication, you may indicate that here to bypass the “Comments to the Author” section, enter your conflict of interest statement in the “Confidential to Editor” section, and submit your "Accept" recommendation.

Reviewer #1: (No Response)

2. Is the manuscript technically sound, and do the data support the conclusions?

Reviewer #1: Yes

3. Has the statistical analysis been performed appropriately and rigorously? 

Reviewer #1: Yes

4. Have the authors made all data underlying the findings in their manuscript fully available?

Reviewer #1: Yes

5. Is the manuscript presented in an intelligible fashion and written in standard English?

Reviewer #1: Yes

6. Review Comments to the Author

Reviewer #1: I am typing these words to complete the 100-character minimum for this item, so that I can submit my recommendation.

7. PLOS authors have the option to publish the peer review history of their article (what does this mean?). If published, this will include your full peer review and any attached files.

Reviewer #1: **Yes: **Thomas A. Stoffregen

---

## [Editor Report · Acceptance letter]

7 Dec 2022

PONE-D-22-03586R2 

Neuromotor changes in participants with a concussion history can be detected with a custom smartphone app 

Dear Dr. Rhea:

I'm pleased to inform you that your manuscript has been deemed suitable for publication in PLOS ONE. Congratulations! Your manuscript is now with our production department. 

Kind regards, 

on behalf of

Dr. Ryan Thomas Roemmich 

Academic Editor

PLOS ONE